# A Deep Reinforcement-Learning Approach for Inverse Kinematics Solution of a High Degree of Freedom Robotic Manipulator

**Aryslan Malik [1,\*][iD], Yevgeniy Lischuk [2], Troy Henderson [1] and Richard Prazenica [1]**

1    Aerospace Engineering Department, Embry—Riddle Aeronautical University, Daytona Beach, FL 32114, USA; hendert5@erau.edu (T.H.); prazenir@erau.edu (R.P.)
2    Software—Device OS, Amazon, Austin, TX 78758, USA; lischuky@amazon.com
\*    Correspondence: malika3@erau.edu

**Abstract:** The foundation and emphasis of robotic manipulator control is Inverse Kinematics (IK). Due to the complexity of derivation, difficulty of computation, and redundancy, traditional IK solutions pose numerous challenges to the operation of a variety of robotic manipulators. This paper develops a Deep Reinforcement Learning (RL) approach for solving the IK problem of a 7-Degree of Freedom (DOF) robotic manipulator using Product of Exponentials (PoE) as a Forward Kinematics (FK) computation tool and the Deep Q-Network (DQN) as an IK solver. The selected approach is architecturally simpler, making it faster and easier to implement, as well as more stable, because it is less sensitive to hyperparameters than continuous action spaces algorithms. The algorithm is designed to produce joint-space trajectories from a given end-effector trajectory. Different network architectures were explored and the output of the DQN was implemented experimentally on a Sawyer robotic arm. The DQN was able to find different trajectories corresponding to a specified Cartesian path of the end-effector. The network agent was able to learn random Bézier and straight-line end-effector trajectories in a reasonable time frame with good accuracy, demonstrating that even though DQN is mainly used in discrete solution spaces, it could be applied to generate joint space trajectories.

**Keywords:** robotics; robotic manipulator; Inverse Kinematics; neural networks; Reinforcement Learning; Deep Learning; Artificial Intelligence; DQN

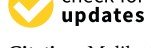



## 1. Introduction

Robotic manipulators' application spans different sectors from automotive to space, where they are used extensively in various industrial manufacturing and assembly processes, including, but not limited to, welding, sorting, and handling radioactive or hazardous materials; surgery; transferring payloads in and out of space stations; etc. As Machine Learning (ML), Artificial Neural Networks (ANNs), and Artificial Intelligence (AI) in general matured in image recognition and vision systems, the next logical step would be to achieve full autonomy of robotic manipulators [1,2]. However, there are certain challenges that have to be overcome to achieve this goal. For instance, depending on the number of joints and links possessed by a manipulator, the mapping from Cartesian space to the robot's joint space becomes very involved, which is problematic, as the tasks that a robotic arm receives are in Cartesian space, whereas the commands (velocity or torque) are in joint space. Therefore, Inverse Kinematics (IK) is arguably one of the most important aspects that has to be addressed if it is desired to have a fully autonomous robotic manipulator.

The solutions to the IK problem can be broadly classified into two categories: pseudo-inverse Jacobian and numerical methods. The analytical solution to IK can be considered as an additional category; however, it is only limited to simple manipulators with a few joints and links, because as the number of joints increases, the robotic arm will tend to have

multiple postures corresponding to a single Cartesian point in space. As such redundancy is introduced, pseudo-inverse Jacobian and numerical methods, such as nonlinear root-finding techniques, are normally employed. The pseudo-inverse Jacobian based methods can be considered to be traditional approaches, but they suffer from an inability to satisfy several objectives at once, for instance, minimizing torque and avoiding obstacles while tracking the desired Cartesian trajectory. The numerical methods, such as the Newton–Raphson nonlinear root-finding technique are, generally, more flexible, but might result in high computational costs or undesired postures, which would require hard coding the joint-space constraints. TRAC-IK is one example of state-of-the-art IK algorithms categorized as numerical methods that employ sequential-quadratic programming (SQP) approaches with a range of quadratic error metrics or even nonlinear constrained optimization [3].

Apart from these categories, the application of Machine Learning (ML), Artificial Intelligence (AI), and Artificial Neural Network (ANN) techniques has gained popularity in recent years. Various RL methods, such as Genetic Algorithm (GA) and Swarm Intelligence (SI) have proven to be effective in solving IK even for robotic manipulators with high-DOFs [4,5]. Deep Deterministic Policy Gradient (DDPG) and Normalized Advantage Function (NAF) algorithms have shown their usefulness in continuous action spaces and, more specifically, in robotic manipulation [6,7]. Continuous-action space approaches, however, suffer from instabilities and a sensitivity to hyperparameters [8–11]. Deep Q-Networks (DQN) proved their usefulness in the robotics field, where their architecture was used for vision-based manipulation [12,13], path planning [14–16], navigation [17,18], and even collision avoidance [19]. However, only a few works have explored ideas of utilizing DQN architecture to solve the IK of high-DOF robotic systems [20–22], and even fewer have implemented the findings on real robotic manipulators.

This work, on the other hand, examines the use of the DQN algorithm for solving the aforementioned IK problem for the 7-DOF robotic manipulator and implements it on a physical Sawyer robotic arm. The DQN algorithm's output encompasses the complete action space of the 7-DOF robotic system, and the IK solution was developed in conjunction with a more recent PoE formulation, which makes this method unique. Despite the fact that the method was tested on a 7-DOF Sawyer robotic arm, it can easily be generalized and applied to any robotic system. Understandably, the DQN algorithm works better for discrete action spaces, and applying it on a continuous action space as joint-space control might seem counterproductive, but this study aims to demonstrate that, with a modest compromise in accuracy, the DQN networks can be trained for generating complex trajectories in joint space even for a high-DOF robotic arm. The application of such methods could be designing joint-space trajectories for a robotic manipulator that perform routine tasks, e.g., space servicing mission or welding in a factory line, which would reduce the input required from the engineers setting up these manipulators, as they would only have to double check the trajectories generated by the network, reducing their workload, therefore saving resources and time. The simpler architecture of the DQN compared to, for instance, DDPG means that it will be easier and faster to tailor it towards specific design objectives. It is also believed that even the compromise in accuracy will be eradicated in the near future, since faster and more reliable hardware becomes more and more readily available, leading the authors to believe that a simpler RL algorithm as DQN will become even more convenient and accessible despite the perceived "curse of dimensionality" [23,24]. Thus, this work aims to present a RL approach based on ANNs for solving the IK problem for robotics with simulations and experimental implementation on the Sawyer 7-DOF robotic manipulator.

The rest of this paper is structured as follows: Section 2 presents materials and methods, Section 2.1 describes the robotic arm and PoE kinematic model employed in this algorithm, and Section 2.2 covers the Deep Q-Network architecture and hyper parameters. Section 3 provides and discusses the simulation results and experimental joint angles, velocities, and torques resulting from algorithm implementation, and, finally, Section 4 concludes the paper with final remarks.

## 2. Materials and Methods

### 2.1. Robotic Arm and PoE-FK Model Description

The 7-DOF Sawyer robotic manipulator by Rethink Robotics was employed for our simulations and experiments. This arm is shown in Figure 1. The Sawyer robotic arm has seven links and seven independently actuated joints. The information regarding the arm's dimensions, weight, inertia properties, joints' positions and limits, etc., was extracted from the Universal Robot Description Format (URDF) file [25] and used in the simulation, where it also served as an input for the RL model.

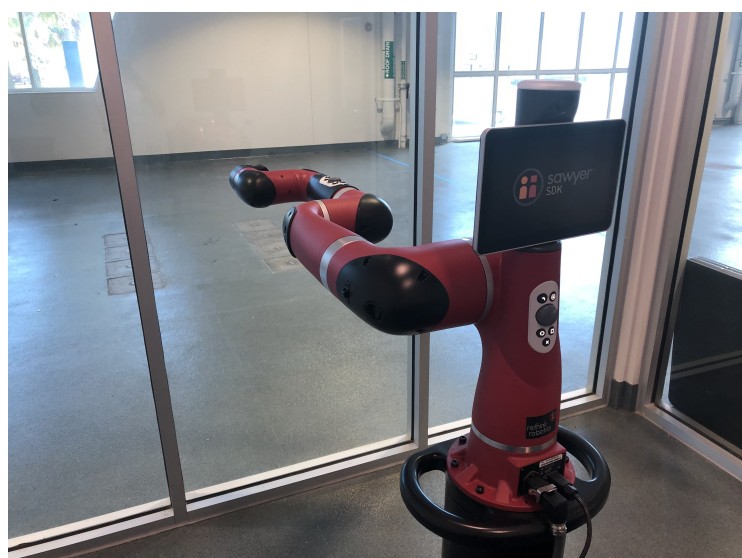

**Figure 1.** Sawyer robotic arm in the home configuration ($\underline{\theta} = 0$) running on SDK mode.

For the FK modeling, the PoE method was employed, as it is easier to configure and does not limit a user to following a strict convention, which is present when working with the more conventional Denavit–Hartenberg (D–H) approach to FK [26]. In order to accurately track the position and orientation of any point on a robotic manipulator, say the end-effector, the PoE-FK model only requires its home configuration and Screw axes represented in inertial (space) frame. The end-effector home configuration is represented as a $4 \times 4$ $SE(3)$ matrix and is shown in Equation (1) with the 7 Screw axes:

$$M_{s7} = \begin{bmatrix} 0 & 0 & 1 & 0.9860 \\ 0 & -1 & 0 & 0.1517 \\ 1 & 0 & 0 & 0.3170 \\ 0 & 0 & 0 & 1 \end{bmatrix} \tag{1}$$

$$\mathcal{S}_1 = \begin{bmatrix} 0 \\ 0 \\ 1 \\ 0 \\ 0 \\ 0 \end{bmatrix} \mathcal{S}_2 = \begin{bmatrix} 0 \\ 1 \\ 0 \\ -0.317 \\ 0 \\ 0.081 \end{bmatrix} \mathcal{S}_3 = \begin{bmatrix} 1 \\ 0 \\ 0 \\ 0 \\ 0.317 \\ -0.1925 \end{bmatrix} \mathcal{S}_4 = \begin{bmatrix} 0 \\ 1 \\ 0 \\ -0.317 \\ 0 \\ 0.481 \end{bmatrix} \tag{2}$$

$$\mathcal{S}_5 = \begin{bmatrix} 1 \\ 0 \\ 0 \\ 0 \\ 0.317 \\ -0.024 \end{bmatrix} \mathcal{S}_6 = \begin{bmatrix} 0 \\ 1 \\ 0 \\ -0.317 \\ 0 \\ 0.881 \end{bmatrix} \mathcal{S}_7 = \begin{bmatrix} 1 \\ 0 \\ 0 \\ 0 \\ 0.317 \\ -0.1603 \end{bmatrix} \tag{3}$$

where the subscript of $M_{s7}$ denotes that it is the home configuration of the 7th joint represented in the inertial frame. The detailed description of the process is outlined in our previous work [27,28]. The Screw axes, given in Equations (2) and (3), are represented in the inertial frame as well. The PoE-FK of the end-effector is thus given as:

$$T_{s7} = e^{[\mathcal{S}_1]\theta_1} e^{[\mathcal{S}_2]\theta_2} \cdots e^{[\mathcal{S}_7]\theta_7} M_{s7} = \begin{bmatrix} R_{s7} & \underline{p}_{s7} \\ 0_{1\times 3} & 1 \end{bmatrix} \in SE(3),$$ (4)

where $(\theta_1, \theta_2, \cdots, \theta_7)$ are the joint positions of the manipulator, and $R_{s7} \in SO(3)$ and $\underline{p}_{s7} \in \mathbb{R}^3$ are the orientation and position of the end-effector, respectively.

Similarly to Equation (4), the PoE-FK representation of any point on a robotic arm can be described if its home configuration is available [29]. The general formula for the PoE-FK of an *i*-th point attached to a *j*-th link can be expressed as follows:

$$T_{si} = e^{[\mathcal{S}_1]\theta_1} e^{[\mathcal{S}_2]\theta_2} \cdots e^{[\mathcal{S}_j]\theta_j} M_{si}$$ (5)

*2.2. Deep Q-Networks for Inverse Kinematics*

2.2.1. Background

DQN belongs to the family of RL algorithms that work by performing continuous iterations over the problem space, collecting knowledge, and selecting the appropriate action. Here, it is used to solve the IK problem by observing the current state of the robot and deciding how the joints should be oriented to achieve the smoothest and the most accurate trajectory. DQN, by itself, can be described as a merger between the Q-Learning and neural networks. It has been shown that it can perform quite challenging and complex tasks, such as mastering Go [30] and several Atari games [31]. Being able to operate in environments with vast solution spaces and state–action pairing, it is of great interest for robotics applications [32].

Similarly to the Q-learning algorithm, the DQN agent operates on the observation, action selection, action execution, and reward obtainment scheme, as defined by the Markov Decision Process (MDP) [33]. This process is essential for the agent's decision making and is described as the tuple $\{\mathcal{S}, \mathcal{A}, \mathcal{R}, \mathcal{T}\}$. Here, $\mathcal{S}$ denotes states of an agent in an environment, $\mathcal{A}$ denotes available actions, $\mathcal{R}$ represents rewards given to an agent, and $\mathcal{T}$ is the transition from one state to another. As for Q-learning, the goal of an agent is to observe the current state and select actions that would maximize the reward. To identify that, Q-learning uses the Q-table that maps states to actions and expected rewards. These expected rewards in the Q-table are specific for each state and are known as Q-values, being calculated using the Bellman equation:

$$Q(s,a) = Q(s,a) + \alpha(R + \gamma \max Q(s',a') - Q(s,a)),$$ (6)

where $Q(s,a)$ is the Q-value with the current state $s$ and action $a$, $\alpha$ is the learning rate of an agent, $R$ is the reward, $\gamma$ is the discount factor, and $Q(s',a')$ is the reward estimation for the next state. The discount factor is important, as it informs the agent how much it should be concerned about future rewards. The low discount factor, such as $\gamma = 0$, would make the agent care about the near future rewards, i.e., rewards that would come in the beginning stages/iterations of the agent's learning process. On the contrary, a larger factor, such as $\gamma = 1$, would make the agent care about the distant future rewards, which denote the rewards that the agent would get during the latter stages/iterations of the training.

When the agent is placed in an environment, it observes the environment by examining the Q-table, chooses actions, performs them, calculates the corresponding Q-value, and updates the Q-table accordingly. It iterates over that process until it reaches the final state, i.e., its end goal. This algorithm is simple to implement and is generally efficient for environment with limited number of states and actions. However, it is not entirely suitable for complex environments, where the number of states is large, or, conversely,

small but continuous [32]. To realize the benefits of Q-learning, but tackle more complex environments, a reinforcement algorithm, such as DQN can be used.

As mentioned earlier, DQN operates in a similar manner to Q-learning; however, the Q-table is replaced with a neural network, with a selected number of nodes and hidden layers, where the weights and biases of these nodes represent Q-values. An additional benefit of DQN lies in a replay buffer, which allows the saving of some experience and the reusing of it during training [33]. This allows the decreasing of training time and increases the stability of an agent. The Bellman equation is adapted for neural networks and represents the squared-loss function, as seen in Equation (7):

$$\mathcal{L}(s, a|\theta) = (R + \gamma \max Q(s', a'|\theta') - Q(s, a|\theta))^2,$$
(7)

where $\theta$ denotes the current weights and biases of the network, $R$ is the reward, and $\gamma$ is the discount factor. Note that the role of the discount factor here is similar to its functionality in Equation (6). Regarding $Q(s', a'|\theta')$ as a whole, it represents the states $s'$, the actions $a'$, and the weights and biases $\theta'$ of the so-called target network, which is a configuration of the network that the agent is trying to achieve during its training. The final parameter, $Q(s, a|\theta)$, is the prediction network. The weights and biases for this network are calculated during each iteration and are then copied to the target network after several more consequent iterations. Note that copying is not performed during each iteration, as it is better for the weights and biases of the target network to remain fixed for several iterations, which leads to a more stable training [33]. After obtaining values for target and predicted networks, the agent performs the gradient descent over the neural network calculating the loss and attempting to minimize it via choosing an appropriate action.

### 2.2.2. Implementation

To implement the DQN agent for solving the IK problem for the 7-DOF robotic manipulator, a simulation was performed using MATLAB with the Deep Learning Toolbox. The benefit of it lies in its quick visualization of the solution as well as the inclusion of all required dependencies. The simulation would provide the trajectory estimates, which would then be used for visualization and could be fed to the real 7-DOF robotic arm. As the proposed algorithm does not involve any constraints on time, the time-scaling can be included after the joint-angle positions are computed to account for joint velocity and torque limits. To perform simulations, a custom environment was created for the agent, where all initial properties were defined for the end-effector frame and joint-space position parameters $\theta$. A total of seven $\theta$ parameters were present, one for each joint; these will serve as states for the DQN agent. This $\underline{\theta}$ vector is the target that is required to be found by an agent for every point along the discretized trajectory via performing the gradient descent over the neural network. After finding and tuning optimal position parameters for one point, the agent would move to the next, compiling all points at the end into the full trajectory matrix. The actions were defined as incremental changes to $\theta$ by a certain amount $\delta\theta = 0.0025°$. The training time is very sensitive to the $\delta\theta$ parameter. In general, a total of three types of action was present: increase $\theta$, decrease $\theta$, or leave it unchanged. As each of these action types can be applied to each joint, the number of all possible combinations equals $3^7 = 2187$. Upon executing an action by updating joint-space position parameters, the end-effector configuration was calculated using the PoE-FK model (Equation (4)). The output of the latter was used to evaluate the fitness of the selected action, which would lead to the reward selection. The fitness itself represents how close the configuration of the end-effector is to the desired configuration on the trajectory. Several fitness functions were examined, with some focusing solely on position error and others evaluating configuration error (position and attitude errors combined). All of the attempted fitness functions are shown in Equations (8)–(12), where the fitness value is a scalar for all cases.

$$f_i^j = \|\underline{p}_{s7}(\theta_i^j) - \underline{p}_d^j\|$$
(8)

$$f_i^j = |\ [1\ 1\ 1]\underline{p}_{s7}(\theta_i^j) - [1\ 1\ 1]\underline{p}_d^j\ |, \tag{9}$$

where $\underline{p}_d^j$ and $\underline{p}_{s7}(\theta_i^j)$ are the (d) desired j-th point's Cartesian coordinates along the trajectory, and the algorithm's i-th iteration of the corresponding point's coordinates, respectively. Fitness functions shown in Equations (8) and (9) are rather simplistic, where only the normal mapping and absolute value operators were used. These functions lead to non-monotonic convergence where the fitness value would oscillate about a certain value close to the goal fitness, $f_g$, and most of the time diverging as the number of iterations increase. Moreover, these fitness functions only take into account the position and do not attempt to minimize the attitude error.

$$f_i^j = \left\| \begin{bmatrix} 0 & 0 & 0 & 1 & 0 & 0 \\ 0 & 0 & 0 & 0 & 1 & 0 \\ 0 & 0 & 0 & 0 & 0 & 1 \end{bmatrix} \log(T_{s7}^{-1}(\theta_i^j)T_{sj})^{-\wedge} \right\| \tag{10}$$

$$f_i^j = \|\log(T_{s7}^{-1}(\theta_i^j)T_{sj})^{-\wedge}\| \tag{11}$$

$$f_i^j = \sigma_p \|\underline{p}_{s7}(\theta_i^j) - \underline{p}_d^j\| + \sigma_R \arccos\left(\frac{tr(R_{s7}(\theta_i^j)R_d^T) - 1}{2}\right), \tag{12}$$

where $T_{sj}$ is the configuration of the desired j-th point along the trajectory, $T_{s7}^{-1}(\theta_i^j)$ is the algorithm's corresponding i-th iteration, log is the matrix logarithm, and the $(-\wedge)$ symbol represents the "un-wedge" mapping such that $se(3) \in \mathbb{R}^{4\times4} \to \mathbb{R}^6$, $\underline{p}_d$, and $R_d$ are desired position and orientation with corresponding weighting parameters, $\sigma_p$ and $\sigma_R$, that are used to prioritize tasks in the workspace. The fitness functions represented in Equations (10)–(12) result in monotonic convergence to the fitness goal value $f_g$ with occasional bounces just around the desired fitness value. Moreover, the fitness function given in Equations (11) and (12) take into account both the position and attitude error, such that the algorithm can be configured to learn to track both the Cartesian trajectory of an end-effector and the attitude. Additional constraints, such as obstacle avoidance, can be incorporated with these fitness functions.

After calculating the fitness, a reward was given to the agent. The reward itself was adjusted dynamically according to the value of the aforementioned fitness. For example, if the fitness of the current iteration after adjusting positions is lower than the fitness for the previous iteration, then the reward is positive, negative if otherwise, and close to no reward if it is exactly the same. To ensure the reward is given in correspondence with how far previous fitness is to the current one, it has been modeled using the arctan function in a range of $(-\pi/2, \pi/2)$ [34]. Doing so allows for avoiding possibly over-rewarding or over-punishing the agent. However, to incorporate the difference between previous and current fitness, as well as the desired goal, the reward is defined as:

$$R = \arctan\left[(f_{i-1}^j - f_i^j)\frac{\pi}{2}\frac{1}{f_g}\right]m, \tag{13}$$

where $f_{i-1}^j$ is the fitness for the previous iteration of the point j, $f_i^j$ is the fitness for the current iteration of the point j, $f_g$ is the goal fitness, and m is the selected magnification factor. The goal fitness represents the fitness value that was manually chosen after some experimentation, it is used to stop calculations for the current point and move to the next, which affects both the accuracy of tracking and the training time. This value was chosen to be $\approx 0.001$. The magnification factor m was another chosen parameter used to increase the output of the arctan, as fitness values tend to be in the order of $10^{-3}$. Based on experimentation, the factor was selected to be $m = 100$. After obtaining the reward, the DQN agent will store its experience in the so-called Experience Buffer. This buffer is also used by the agent to sample some small batches of the $\{\mathcal{S}, \mathcal{A}, \mathcal{R}, \mathcal{T}\}$ tuple to essentially replay some experiences, allowing it to learn more from the same data. Lastly, the agent will calculate the overall network loss using Equation (7) and update the weights and biases

of the network accordingly. The general algorithm for how the DQN agent operates in an environment for finding the joint-space trajectory is presented in Algorithm 1.

---

**Algorithm 1** Pseudo-code for finding joint positions on the trajectory using DQN.

---

Initialize custom simulated training environment for the DQN agent.
Set the initial point of the screw axis to be the closest to the Bézier curve.
Initialize the DQN Critic network.
Reset initial states: joint positions $\{\theta_1, \theta_2, \theta_3, \theta_4, \theta_5, \theta_6, \theta_7\}$ for the TrajectoryPoint $j = 1$.

**for** TrajectoryPoint $j = 2$ to MaxTrajectoryPoints **do**

    **for** Episode $i = 1$ to MaxEpisodes **do**
        Observe states $\{\theta_1, \theta_2, \theta_3, \theta_4, \theta_5, \theta_6, \theta_7\}$ for TrajectoryPoint $j - 1$ during Episode $i$.
        Select action based on DQN Critic network's weights and the exploration factor.
        Update states $\{\theta_1, \theta_2, \theta_3, \theta_4, \theta_5, \theta_6, \theta_7\}$ by the amount $\delta\theta$ according to the selected
action.
        Perform action via:
$$T_{si} = e^{[\mathcal{S}_1]\theta_1} e^{[\mathcal{S}_2]\theta_2} \cdots e^{[\mathcal{S}_j]\theta_j} M_{si}$$
        Evaluate fitness:
$$f_i^j = \sigma_p \|\underline{p}_{s7}(\theta_i^j) - \underline{p}_d^j\| + \sigma_R \arccos\left(\frac{tr(R_{s7}(\theta_i^j)R_d^T)-1}{2}\right)$$
        Obtain the reward:
$$R = \arctan\left((f_{i-1}^j - f_i^j)\frac{\pi}{2}\frac{1}{f_g}\right)m$$
        Store experience in the ExperienceBuffer.
        Sample random minibatch of transitions from the ExperienceBuffer.
        Calculate DQN Critic network loss $\mathcal{L}$ and update network weights.
        **if** Fitness == Goal **then**
            *break*
        **end if**
    **end for**
**end for**

---

The model architecture consists of four main layers: an input layer, two hidden layers, and an output layer. As was mentioned earlier, seven joint-space positions, $\theta$, served as states, and, architecturally speaking, inputs to the neural network. It is followed by a first fully connected, or, in other words, a dense layer with 128 nodes accompanied by a Rectified Linear Unit (ReLU) activation function. This function was chosen due to its computational efficiency and general popularity. Next, another fully connected layer was included with 256 nodes and another ReLU function. Finally, the output layer represents actions with a total of 2187 nodes, one for each action type available for each joint. This architecture can be seen in Figure 2. For a summary of some other parameters, such as the learning rate, discount factor, and gradient threshold, as well as the selected parameters for the reward calculation per Equation (13), view Table 1.

**Table 1.** Summary of selected parameters for the DQN agent and parameters for the reward function calculation.

| Parameter | Value |
|---|---|
| Joint-Space Position Step $\delta\theta$ | $\pm 0.0025°$ |
| Learning Rate | 0.01 |
| Discount Factor | 0.9 |
| Gradient Threshold | 1 |
| Mini Batch Size | 64 |
| Target Smooth Factor | $1 \times 10^{-3}$ |
| Goal Fitness $f_g$ | 0.001 |
| Magnification Factor $m$ | 100 |

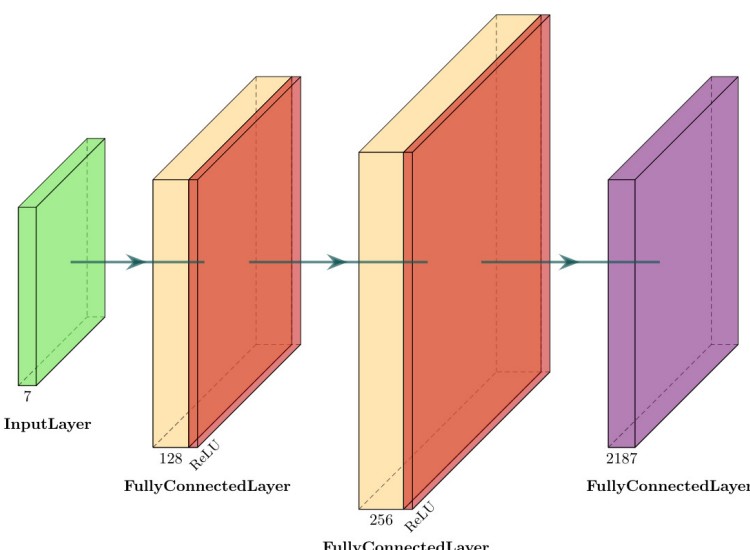

7
InputLayer

128 ReLU
FullyConnectedLayer

256 ReLU
FullyConnectedLayer

2187
FullyConnectedLayer

**Figure 2.** DQN model used for solving Inverse Kinematics corresponding to a random Bézier trajectory of the robotic manipulator.

### 3. Results and Discussion

A workstation with an Intel® Core™ i5-10600K Processor, 16 GBs of RAM, was used to host the simulation of the DQN algorithm for solving IK in MATLAB. For the sake of brevity, this DQN agent, along with the IK computation and solution, can be referred to as DQN-IK. Random Bézier curves were generated and provided as an input to the simulation, and, on this workstation, it takes this algorithm on average 30 s to solve for one Cartesian point. Straight-line Cartesian trajectories were also tested to show that the DQN-IK agent can be generalized to a wide range of end-effector trajectories. The DQN-IK is a bit slow at the beginning of the trajectory for the first 3–4 points but then speeds up for the rest, which is a result of the aforementioned Experience Buffer. The hyper-parameters presented in Table 1 were obtained by rigorous studies and performance comparisons. The goal fitness value of $f_g = 0.001$ led to a maximum allowable error in end-effector position of 1 mm. The goal fitness value can be increased to speed up the learning of the DQN agent but would lead to inaccurate positioning of the end-effector and, vice versa, the smaller the goal fitness, the more accurate the positional tracking is, and the longer it takes for the agent to learn. The architecture of the neural network (Figure 2) was also tweaked to strike a balance between the complexity and the speed of the algorithm. As was mentioned in Section 2.2, the number of hidden layers was chosen to be two with 128 and 256 nodes, respectively. This number of nodes was determined from experimentation and found to provide a good balance between the accuracy of the found solution and the computation speed. Higher number of nodes as well as layers did not provide sufficient gains. Lower number of nodes, e.g., 64 and 128, can be used as well, but at a minor cost to accuracy. The number of hidden layers is significant, because a single-layer design is insufficient, rendering the DQN agent incapable of finding a solution; hence, two layers provide a reasonable implementation.

As there are infinite solutions in the joint space, the DQN-IK algorithm was able to generate several different joint-space trajectories corresponding to a single Bézier curve path. This is demonstrated in Figure 3, where it is evident that the algorithm approached the same path with different solutions. The heat-maps were different for the same path as well. A sample heat-map is shown in Figure 4, and a corresponding bar chart is shown in Figure 5, where it can be seen that, in this particular case, the algorithm has utilized a couple of distinct actions more than the others.



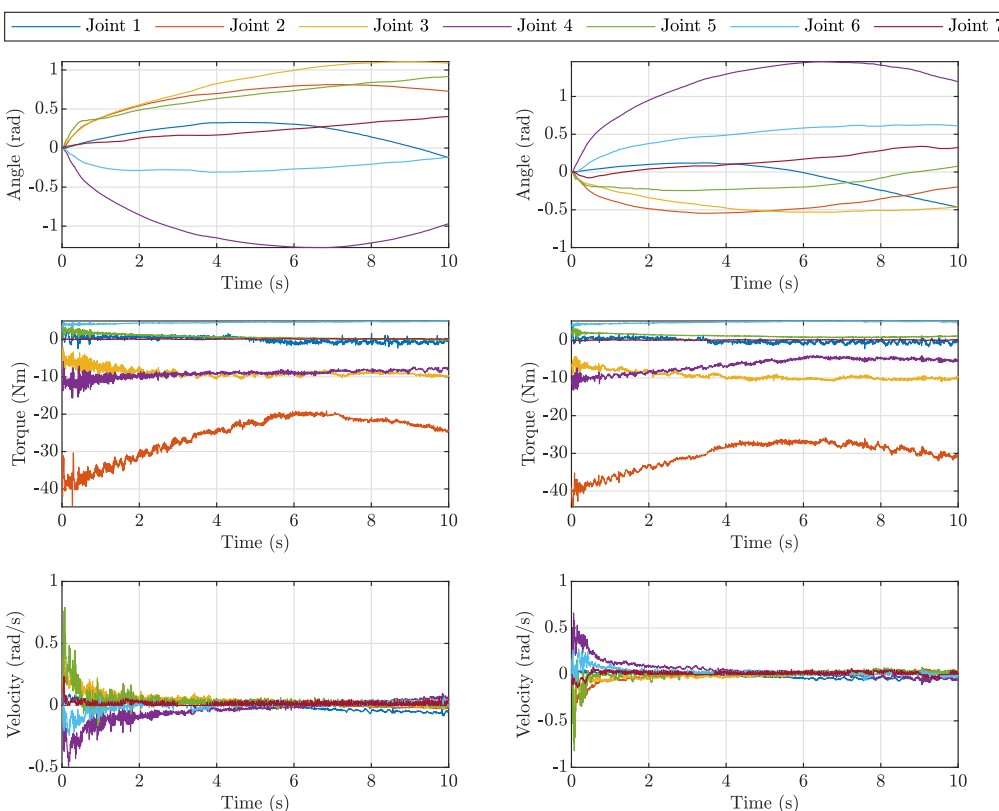

**Figure 3.** Experimental trajectories recorded by the Sawyer robotic arm: two different (**left** and **right** columns) IK solutions to the same Bézier curve Cartesian path produced by DQN-IK.

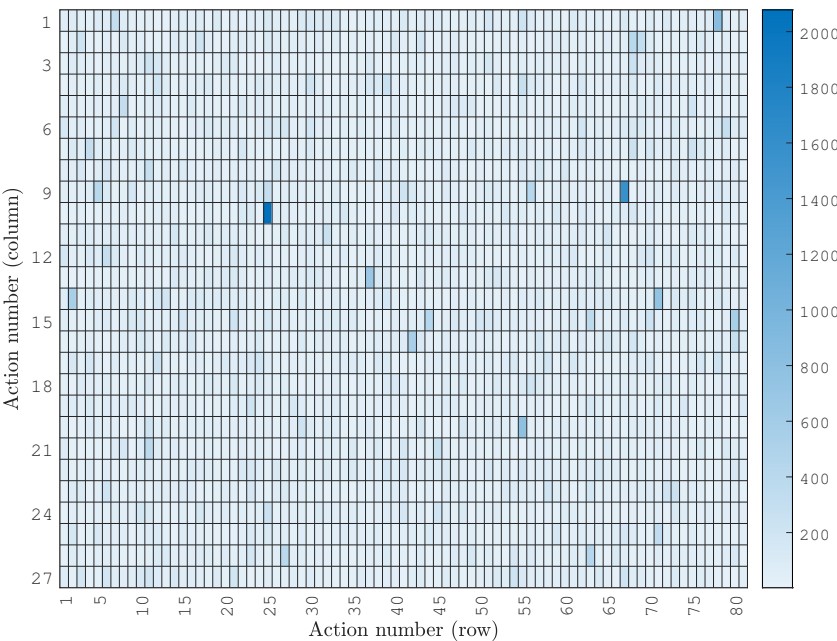

**Figure 4.** Heat-map corresponding to a DQN-IK trajectory.

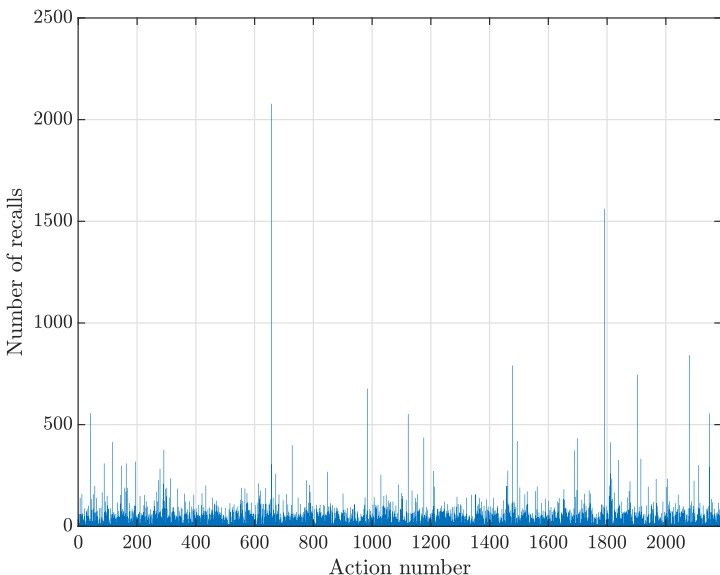

**Figure 5.** Bar chart of the DQN-IK trajectory.

After the curve-path calculation, the algorithm was evaluated based on properties such as achieved fitness, training loss, and cumulative reward. These properties were examined for each point on the calculated trajectory, to see how the algorithm performed. Plots for such analysis of the Bézier curve can be seen in Figure 6. It must be noted that even though a hundred points were used in the curve calculation, only several points is enough to see the general behavior of the algorithm. In addition to that, another trajectory was used in analysis. This trajectory was a straight line, and it was mainly used to compare performance, loss, reward, and fitness between the two trajectory calculations, as well as to present the ability of the algorithm to compute different types of trajectories. Plots for the analysis of this trajectory are shown in Figure 7.

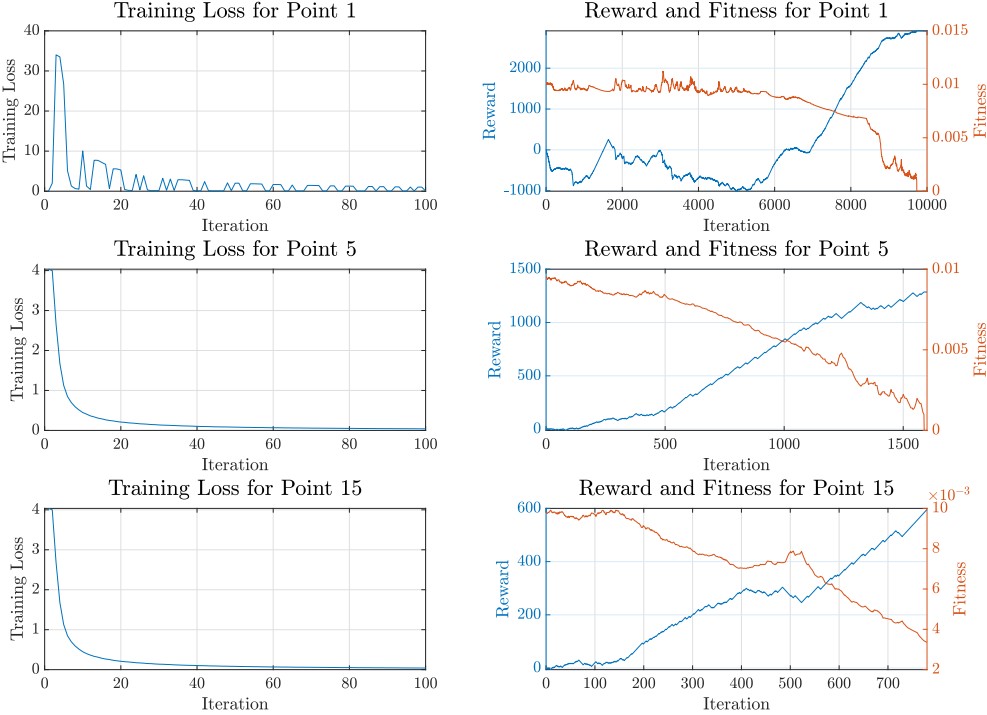

**Figure 6.** Loss, Cumulative Reward, and Fitness for points 1, 5, and 15 of the curved trajectory calculation, respectively.

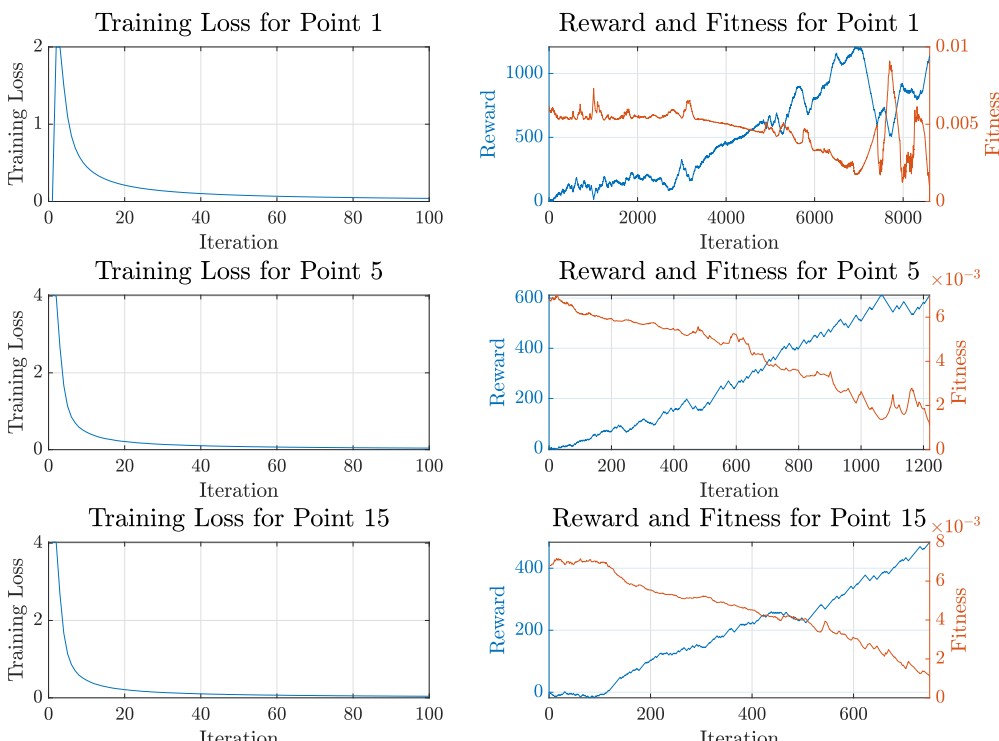

**Figure 7.** Loss, Cumulative Reward, and Fitness for points 1, 5, and 15 of the straight line trajectory calculation, respectively.

Regarding the evaluated properties, in Figures 6 and 7, training loss, reward, and fitness are presented only for points 1, 5, and 15, because it shows the general trend of the training capturing the initial dynamics. The training loss converged for all points demonstrating that the training was successful. It takes about 10,000 and 8000 iterations in Figures 6 and 7, respectively, for the agent to solve for the first point. DQN-IK is relatively slow at the beginning, because it is practically "deciding" which way to proceed, as there are no additional constraints on the joint space. Once the general direction of the joint movement is set, the agent performs better which results in less iterations per Cartesian point along the trajectory, as demonstrated in Figure 6, where point 5 requires about 1500 iterations, and point 15 requires approximately 750. A similar trend can be observed with the straight-line trajectory in Figure 7, where point 5 needed about 1200 iterations, and point 15 only 800.

The inverse correlation between the reward and fitness is also evident from Figures 6 and 7, as expected from Equations (12) and (13). In some cases, the DQN-IK agent arrives at local minimum fitness, e.g., reward and fitness for point 1 in Figure 7 at iteration 7000, which forces it to explore the discretized solution space reducing the reward and increasing the fitness at first with further recovery, improved fitness, and increased reward. This shows the significance of the chosen reward function—as over-rewarding or over-punishment may lead to limited exploration or repeated exploitation, leading to undesired results.

The experimental implementation was carried out by sending the joint trajectory data to the Sawyer robotic arm through the Ubuntu-ROS-Sawyer framework, where the commands were provided as an input for the Python-based controller and sent to the robotic arm in the Software Development Kit (SDK) mode. The maneuver-execution time was selected to be large enough ($t_f = 10$ s) to avoid saturation of the joint limits, since the algorithm does not solve for a time-optimal solution. The time scaling was considered to be linear for the experiments; however, higher-order time scalings can be used to reduce jerk, hence reducing vibration within the robotic system, based on the joint velocity limit constraint. Joint angle positions, velocities, and efforts (torques) were recorded and are shown in the Figure 3. The recorded experimental joint-space trajectories are almost

indistinguishable from the DQN-generated profiles. In Figure 8, it is demonstrated that the Cartesian error stays within the bounds of ±1 mm, except the initial jolt where the error spikes up to 2 mm for a fraction of a second. Figure 9 demonstrates that the tracking error for the straight-line path does not exceed 1 mm. The resulting end-effector Cartesian path is demonstrated in Figure 10. As shown in this figure, the algorithm has found different solutions; in the first one, the manipulator tracks the trajectory from "below" (left column) and in the second from "above" (right column). The DQN-IK successfully generated a straight-line trajectory as shown in Figure 11, with the desired Cartesian path indicated using a green line. The DQN-IK algorithm can also be utilized for generating trajectories avoiding obstacles in the task space. To perform this, the fitness function should be modified to take into account the collision condition, which could be manifested by a large increment in the fitness. The collision condition can be tracked using the PoE-FK model, shown in Equation (4). Generally, the outlined algorithm is very versatile and modular, as various constraints can be incorporated to meet different design objectives.

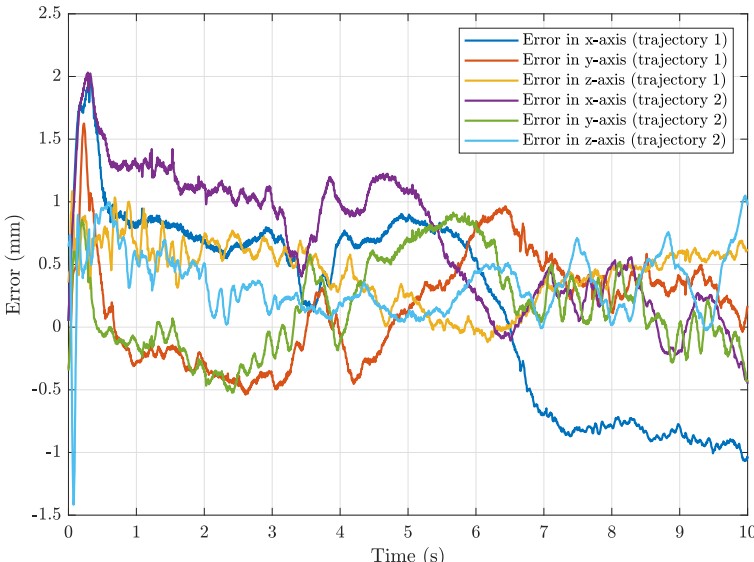

**Figure 8.** Error in Cartesian components of the end-effector during Bézier curve tracking, desired path subtracted from experimental data.

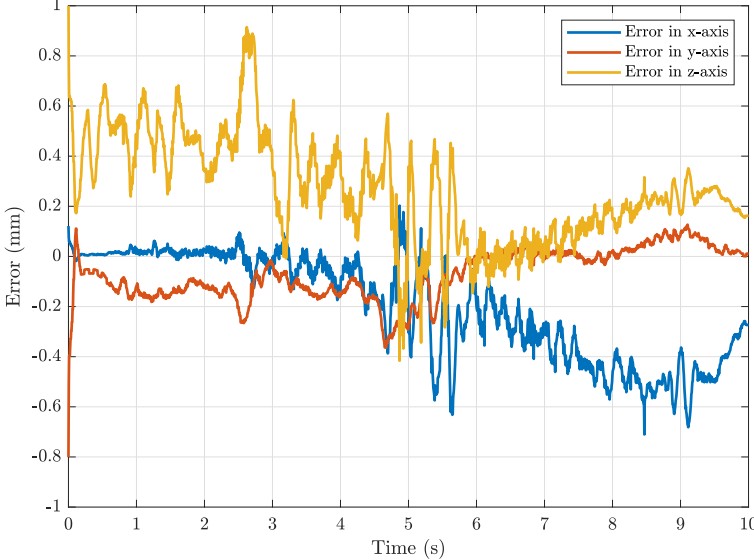

**Figure 9.** Error in Cartesian components of the end-effector during straight-line tracking, desired path subtracted from experimental data.

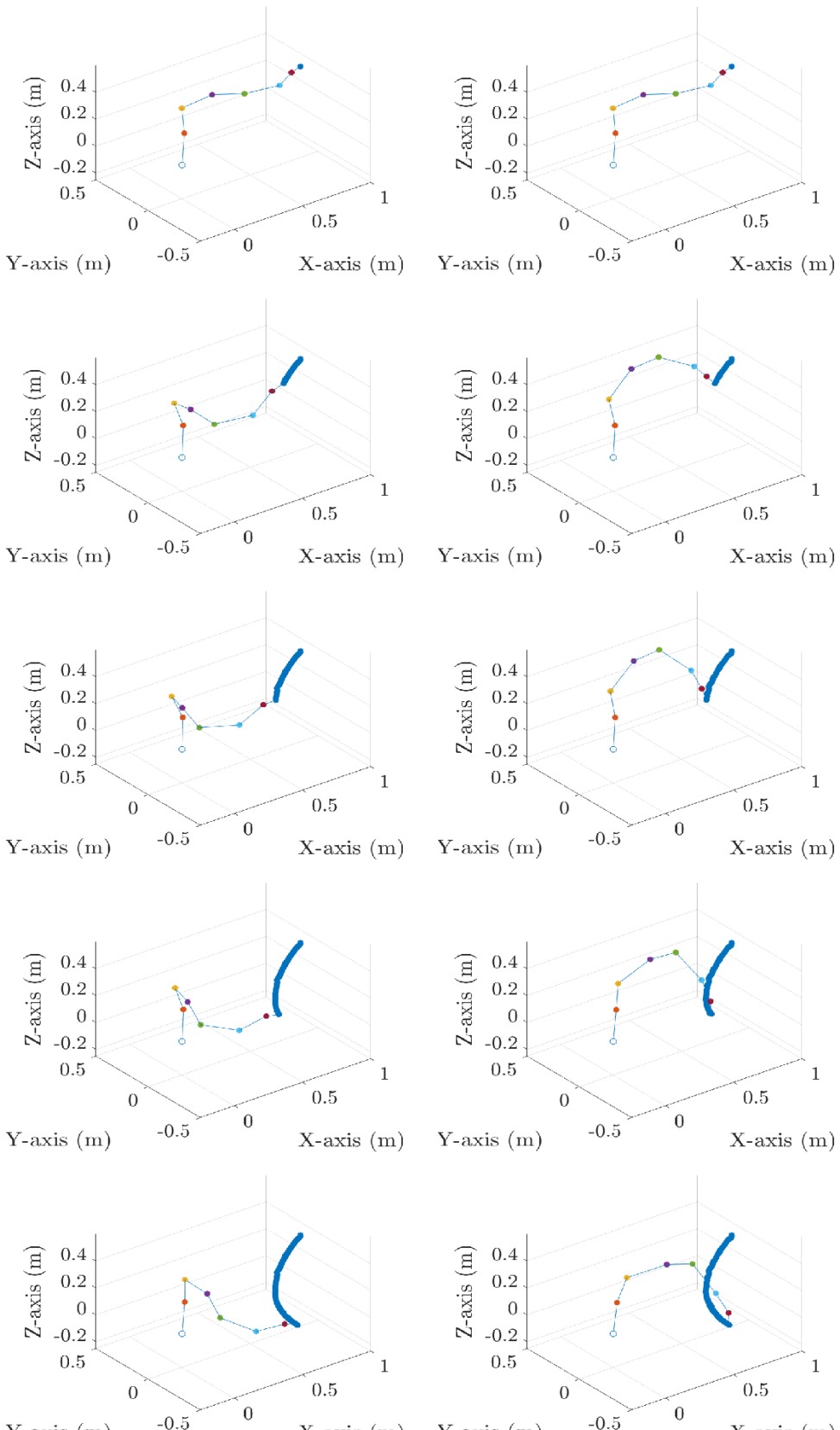

**Figure 10.** Two IK solutions for the same Bézier curve path produced by DQN-IK.

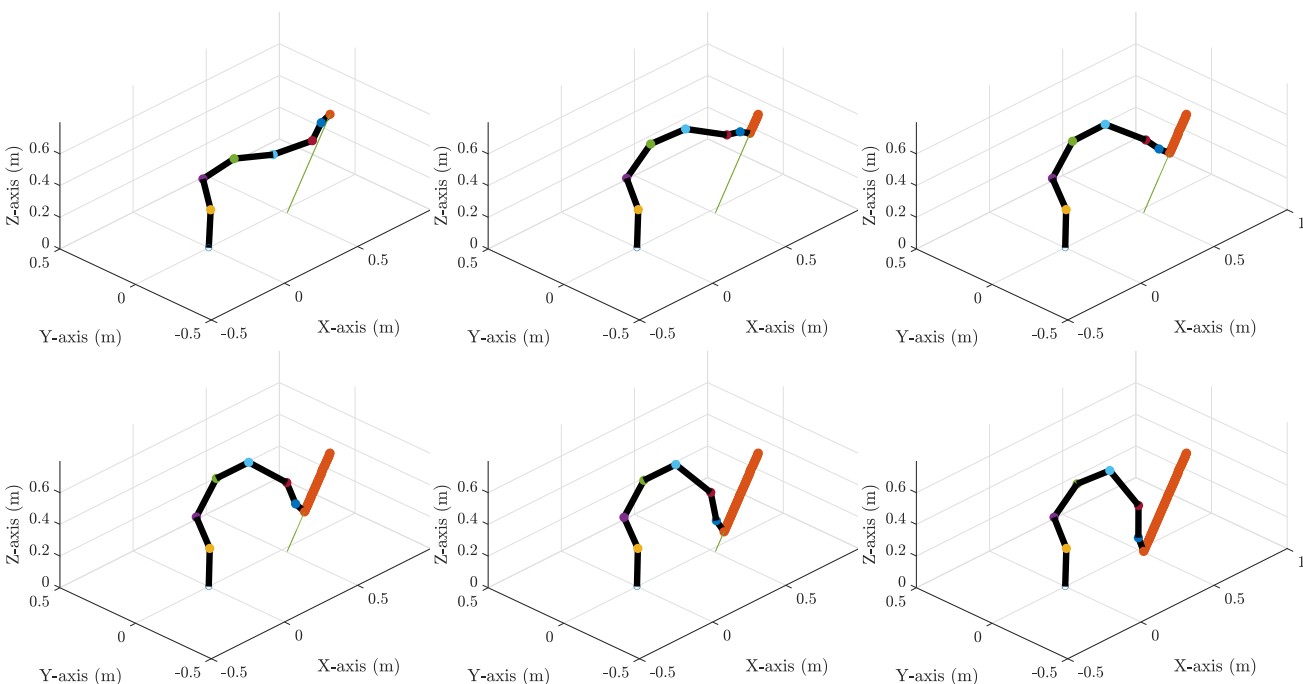

**Figure 11.** Sawyer robotic arm IK solution for straight-line path produced by DQN-IK. Green line represents the desired end-effector path, and orange line represents the actual path.

## 4. Conclusions

In this work, it was shown that the Deep Reinforcement Learning approach using DQN is viable for solving the Inverse Kinematics problem of a high-DOF robotic manipulator. The description of the robotic arm and the PoE-FK model was presented. In general, home configuration and Screw axes, which are constant vectors, are all that is necessary to track the position and/or orientation of any point or joint of the robotic manipulator. The proposed approach was used in conjunction with the DQN algorithm to calculate the optimal joint positions to achieve the required trajectory. The DQN-IK algorithm takes in joint-space position parameters as inputs and selects an appropriate action to adjust them in order to track Cartesian points on the desired trajectory. The algorithm itself is comprised of the neural network with four layers: input, two hidden layers, and one output layer. During operation and for each iteration, the DQN agent attempts to select the appropriate action in terms of increasing/decreasing certain joint-space position values by a specified amount. The algorithm is then evaluated on its action via the calculation of the fitness value. The aim of this fitness is to calculate and demonstrate how close the posture described by all seven joint positions is to the desired point on the Cartesian trajectory. This fitness is compared to the value of the previously calculated fitness, and the reward is given accordingly. The reward is dynamically adjusted based on how far the previous fitness is from the newly calculated one. This allows for the guiding of the DQN agent more closely to its goal, decreasing the computation time without giving too much reward/punishment to the agent while trying to determine and optimize joint positions and to stabilize the convergence. The results were also analyzed based on properties such as the training loss, the cumulative reward, and the achieved fitness. It was shown that the DQN-IK agent has some uncertainty when calculating several initial points on the trajectory as indicated by higher loss and fitness and takes more iterations to reach the desired fitness. This, of course, is relative to the consequent trajectory points where the algorithm has a much lower loss as well as fitness. This is explained by the agent not "knowing" the general direction of the desired motion and, thus, performing an exploration of the solution space. It was also demonstrated that the agent is able to compute different types of trajectories, as was presented via both the Bézier curve and a straight-line calculation. Furthermore, the

algorithm can be generalized to other robotic systems, as it will only require modifying the home configuration matrix and Screw axes.

The movement of the 7-DOF robotic arm was simulated using MATLAB and then, using generated data, was executed on the Sawyer robotic manipulator. During the simulation, it was observed that, due to the infinite number of solutions in the joint space, the DQN algorithm can find new trajectories each time it runs. It was demonstrated that, even though DQN, by itself, is bound to using discrete action spaces, using DQN for Inverse Kinematics proved to be a viable option, especially considering that its architecture is less complicated than DDPG or NAF. The more elegant approach may still lie in using DDPG, as its action spaces are continuous. Nevertheless, while being more simple architecture-wise than DDPG, DQN proved to provide a solution with good accuracy in a reasonable computation time and can be modified to fit different needs and objectives. Additionally, DQN avoids a potential pitfall that may exist for DDPG, as some instabilities may be introduced because continuous action spaces are more sensitive to hyperparameters. This is in contrast to discrete action spaces utilized by DQN. Regarding the DQN-IK agent described in this work, its neural network parameters that are shown in this paper were selected after extensive testing and optimization but can be further adjusted to optimize the algorithm and decrease the computation time for a specific design objective.

**Author Contributions:** Conceptualization, A.M., Y.L., T.H. and R.P.; data curation, A.M. and Y.L.; formal analysis, A.M., Y.L., T.H. and R.P.; investigation, A.M. and Y.L.; methodology, A.M. and Y.L.; project administration, T.H. and R.P.; resources, T.H. and R.P.; software, A.M. and Y.L.; supervision, T.H. and R.P.; validation, A.M. and Y.L.; visualization, A.M. and Y.L.; writing—original draft, A.M. and Y.L.; writing—review and editing, A.M., Y.L., T.H. and R.P. All authors have read and agreed to the published version of the manuscript.

**Funding:** This research received no external funding.

**Institutional Review Board Statement:** Not applicable.

**Informed Consent Statement:** Not applicable.

**Data Availability Statement:** Raw data were generated at MicaPlex, Embry-Riddle Aeronautical University. Derived data supporting the findings of this study are available from the corresponding author A.M. on request.

**Acknowledgments:** The authors would like to thank Daniel Posada for his technical support with setting up the Ubuntu-ROS-Sawyer framework.

**Conflicts of Interest:** The authors declare no conflict of interest.

## Abbreviations

The following abbreviations are used in this manuscript:

| | |
|---|---|
| AI | Artificial Intelligence |
| ANN(s) | Artificial Neural Network(s) |
| CG | Center of Gravity |
| DDPG | Deep Deterministic Policy Gradient |
| DoF | Degree of Freedom |
| DQN | Deep Q-Network |
| D–H | Denavit–Hartenberg |
| FK | Forward Kinematics |
| GA | Genetic Algorithm |
| IK | Inverse Kinematics |
| MDP | Markov Decision Process |
| ML | Machine Learning |
| NAF | Normalized Advantage Function |
| QPFJ | Quintic Polynomial Finite Jerk |
| RL | Reinforcement Learning |
| ROS | Robot Operating System |

| SI | Swarm Intelligence |
|---|---|
| SQP | Sequential-Quadratic Programming |
| PoE | Product of Exponentials |
| PSO | Particle Swarm Optimization |
| URDF | Universal Robot Description Format |

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
