# Peer review of "A Deep Reinforcement-Learning Approach for Inverse Kinematics Solution of a High Degree of Freedom Robotic Manipulator"

_robotics, doi:10.3390/robotics11020044_

Round 1

Reviewer 1 Report

This paper mainly studies an inverse kinematics solution method of the high degree of freedom robots combining deep reinforcement learning and PoE,
which needs to be improved in the following aspects:
1. There is an error in Eq. 4.
2. On page 5, line 145, the meaning of gamma is incomplete.
3. Are there any misinterpretations of $Q(s',a'|\theta ') and Q(s,a|\theta)$ on lines 146 and  147?
4. What does the horizontal axis of Fig. 3 and Fig. 4 represent? It should be explained.
5. Figure 10 shows two sequences of solutions of configurations corresponding to the given Bézier trajectory, but it can be seen from the last two subgraphs of Fig. 10 that the corresponding final pose is inconsistent. Does this Bézier trajectory contain attitude information? 
6. For the method proposed in this paper, how to ensure the consistency of the results of the inverse kinematic solution of the high DOF robot? That is, the results of each run are the same. I think deterministic results are more useful to the industry.
7. Reference 25 is incomplete.

Reviewer 2 Report

The paper investigates a deep reinforcement learning approach for solving of inverse kinematics for 7-dof industrial robot. The topic of the paper is interesting, however, I have several issues which should be solved.

Here are my suggestions and comments:

  • Abstract: you didn't describe any background of the investigated topic at the beginning of the abstract.
  • Introduction: An introduction is on the one hand correct, but on the other hand, it doesn't offer exactly described novelty of the paper. There is written that it will be used DQN for solving of IK solution, however, what is the novelty ? Your approach seems to be common. This point is really important. Now , it is hard to say if your work brings any contributions.

Round 2

Reviewer 1 Report

The authors have solved the previous problems, and I agree with them.

Reviewer 2 Report

ok